# Endotaxial stabilization of 2D charge density waves with long-range order

Suk Hyun Sung [1], Nishkarsh Agarwal [1], Ismail El Baggari[2], Patrick Kezer [3], Yin Min Goh [4], Noah Schnitzer[5,6], Jeremy M. Shen[3], Tony Chiang [1], Yu Liu [7], Wenjian Lu [7], Yuping Sun [7,8,9], Lena F. Kourkoutis [6,10], John T. Heron [1,11], Kai Sun [12] & Robert Hovden [1,11] ✉

Charge density waves are emergent quantum states that spontaneously reduce crystal symmetry, drive metal-insulator transitions, and precede super-conductivity. In low-dimensions, distinct quantum states arise, however, thermal fluctuations and external disorder destroy long-range order. Here we stabilize ordered two-dimensional (2D) charge density waves through endo-taxial synthesis of confined monolayers of 1T-TaS$_2$. Specifically, an ordered incommensurate charge density wave (oIC-CDW) is realized in 2D with dra-matically enhanced amplitude and resistivity. By enhancing CDW order, the hexatic nature of charge density waves becomes observable. Upon heating via in-situ TEM, the CDW continuously melts in a reversible hexatic process wherein topological defects form in the charge density wave. From these results, new regimes of the CDW phase diagram for 1T-TaS$_2$ are derived and consistent with the predicted emergence of vestigial quantum order.

Some exotic crystals spontaneously reorganize their valence electrons into periodic structures known as charge density waves (CDWs). In essence, two crystals emerge—the underlying atomic lattice and the emergent charge lattice. Just like atomic crystals, a charge density wave has defects: dislocations, disclinations, and elastic deformation[1–3]. Furthermore, the charge density wave can undergo phase transitions wherein the charge lattice unit cell changes shape and size. All of this CDW reshaping and topological restructuring occurs even when the underlying atomic lattice remains unchanged.

In low dimensions, these quantum phase transitions are promis-ing candidates for novel devices[4–7], efficient ultrafast non-volatile switching[8–10], and suggest elusive chiral superconductivity[11–13]. Unfor-tunately, 2D CDWs are inherently unstable and accessing low-dimensional CDWs remains a challenge[14–16]. Even worse, at elevated temperatures where devices typically operate, disruption of charge density waves is all but guaranteed due to ever-present disorder[17–19]. A long-range ordered incommensurate CDW has yet to be reported.

Here we stabilize ordered incommensurate charge density waves (oIC-CDW) at elevated temperatures ($T_{IC}$ = 350 K) in two dimensions by endotaxial synthesis of TaS$_2$ polytype heterostructures. The estimated hundred-fold amplitude enhancement of the in-plane charge density wave has an increased coherence length comparable to the underlying atomic crystal. The enhanced order of the oIC-CDW increases electro-nic resistivity. This substantial enhancement of charge order is achieved through encapsulation of an isolated octahedral TaS$_2$ CDW layer within a matrix of prismatic TaS$_2$ metallic layers via 2D endotaxial synthesis.

Realizing the ordered incommensurate CDW reveals CDWs have hexatic structure at high-temperature—that is, long-range translational

[1]Department of Materials Science and Engineering, University of Michigan, Ann Arbor, MI 48109, USA. [2]Rowland Institute at Harvard, Cambridge, MA 02142, USA. [3]Department of Electrical and Computer Engineering, University of Michigan, Ann Arbor, MI 48109, USA. [4]John A. Paulson School of Engineering and Applied Sciences, Harvard University, Cambridge, MA 02138, USA. [5]Department of Materials Science and Engineering, Cornell University, Ithaca, NY 14853, USA. [6]Kavli Institute at Cornell for Nanoscale Science, Ithaca, NY 14853, USA. [7]Key Laboratory of Materials Physics, Institute of Solid State Physics, Chinese Academy of Sciences, Hefei 230031, PR China. [8]Collaborative Innovation Centre of Advanced Microstructures, Nanjing University, Nanjing 210093, PR China. [9]High Magnetic Field Laboratory, Chinese Academy of Sciences, Hefei 230031, PR China. [10]School of Applied and Engineering Physics, Cornell University, Ithaca, NY 14853, USA. [11]Applied Physics Program, University of Michigan, Ann Arbor, MI 48109, USA. [12]Department of Physics, University of Michigan, Ann Arbor, MI 48109, USA. ✉e-mail: hovden@umich.edu

symmetry is limited by proliferation of topological defects (i.e., dislocations and disclinations) in CDWs. We show at high temperatures, the CDWs in TaS$_2$ continuously melt as additional dislocations and disclinations form in the charge lattice. This hexatic CDW melting process was not previously observable since the incommensurate CDW normally emerges as a highly-disordered, melted state. By restoring in-plane order through 2D endotaxy, we can reversibly melt and unmelt CDWs in TaS$_2$. Based on these results, we access new regimes of the CDW phase diagram for octahedrally coordinated TaS$_2$ in temperature vs. disorder space. Similar vestigial ordering (i.e., hexaticity) was predicted by Nie, Tarjus and Kivelson[18]; however, with 2D endotaxy we can now tune down the disorder in the CDW phase diagram.

## Results

### The ordered incommensurate charge density wave

The ordered incommensurate CDW (oIC) reported herein (Fig. 1a–d) is strikingly distinct from the well-known incommensurate (IC) CDW (Fig. 1e–h) found in 1T-TaS$_2$ or 1T-TaSe$_2$. Here, the oIC phase is a truly two-dimensional (2D) CDW with long-range positional and orientational order that couples strongly with the underlying crystal lattice (Fig. 1a). The oIC-CDW, illustrated in Fig. 1b, is a crystalline charge-lattice with well-defined, sharp peaks in Fourier space (Fig. 1b-inset). This CDW charge lattice ($a_{CDW} = 11.87$ nm) exists within an underlying atomic lattice illustrated in Fig. 1c.

Electron–lattice interaction is an essential aspect of CDWs, and associated soft-phonon modes manifest as static periodic lattice distortions (PLDs) that reduce crystal symmetry and lower the electronic energy[20,21]. For TaS$_2$, the CDW pulls atoms toward the nearest charge maximum to form periodic clusters of atoms (Fig. 1c, see also Supplementary Fig. 1). Notably for incommensurate charge ordering, each cluster is distinct since the atomic lattice is not commensurate with the CDW[22]. While these lattice distortions are small (<10 pm), selected area electron diffraction (SAED) is sensitive to subtle picoscale distortions and making it a popular choice for characterization of CDW/PLDs[23]. CDW/PLDs diffract incident swift electrons into distinct superlattice peaks decorating each Bragg peak[2,24–26]. In reciprocal space, the CDW charge lattice (Fig. 1b-inset) and the measurable atomic superlattice peaks (Fig. 1c-inset) have corresponding spacing, symmetry, and intensity.

Diffracted superlattice peaks provide a direct measure of the CDW lattice and contain rich information on their order-disorder. Specifically, diffraction represents an ensemble average of the structure over the selected area, and disorder manifests as diffused diffraction peaks[27,28]. Disorder of CDWs smears superlattice peaks but leaves the principal Bragg peaks unaffected (Fig. 1g-inset). For oIC-CDWs, the charge lattice is ordered with limited defects, thus diffraction shows both sharp superlattice and Bragg peaks (Fig. 1c-inset). In contrast, the well-known IC-CDW in 1T-TaS$_2$ possesses significant disorder of its charge distribution. Across decades, the IC phase in 1T-TaS$_2$ is reported with a ring-like, azimuthally diffuse diffraction around each Bragg peak[24,29–31], yet the origin of the diffused superlattice peaks is hardly discussed[32,33].

Here, we present the well-known IC-CDW in bulk 1T-TaS$_2$ as a hexatically disordered charge lattice containing dislocations and disclinations (Fig. 1f). In-situ SAED of 1T-TaS$_2$ taken at 408 K (Fig. 2a) shows azimuthally blurred first order superlattice peaks (marked brown). Averaging all six third-order Bragg peaks (inset, $\Gamma_3$) better highlights this point. Notably, hexatic phases are known to have six-fold rotationally symmetric, azimuthally diffused peaks[34]. The experimental diffraction of IC-CDWs are consistent with a hexatic charge distribution (Fig. 1f)[28,33–35] and corresponding azimuthally diffuse structure factor (Fig. 1f, g-inset). The IC-CDWs are three-dimensional (or quasi-2D) with non-negligible out-of-plane interactions (Fig. 1e–h).

In contrast, the oIC-CDW shows drastically sharper and stronger superlattice peaks measured by in-situ SAED at 408 K (Fig. 2b).

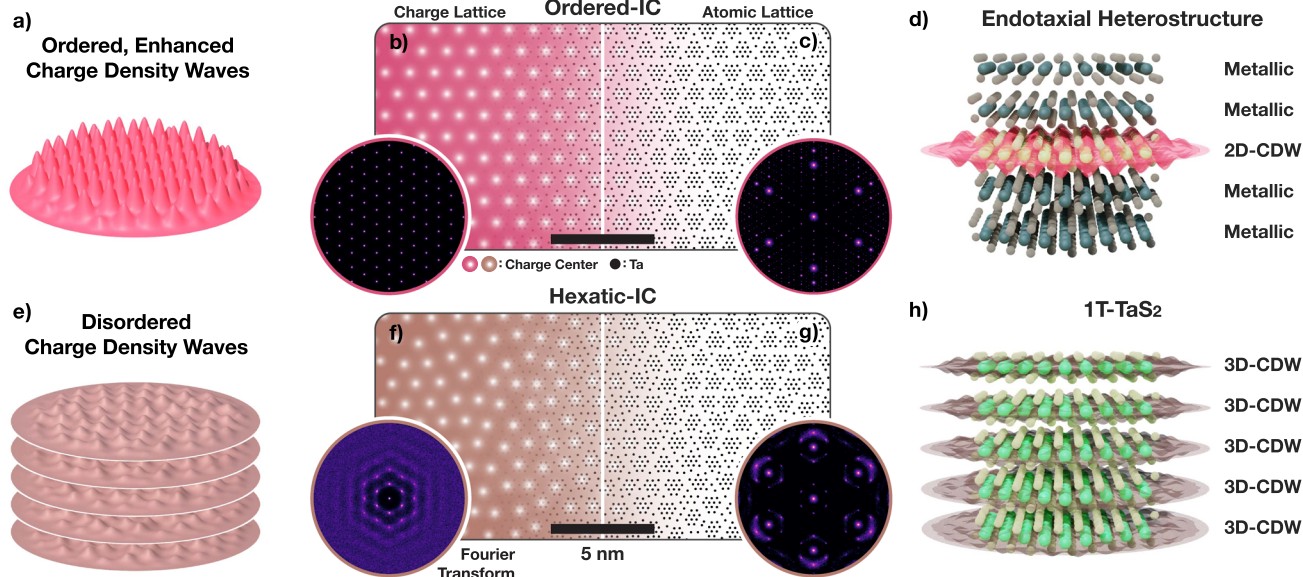

**Fig. 1 | Long-range ordered incommensurate charge density waves. a** Schematic representation of ordered IC-CDW. The CDW is two-dimensional with little disorder. **b** Ordered IC-CDW illustrated as a crystalline charge-density lattice. Here, white spots represent charge centers. Inset) Fourier transform of the charge lattice shows well-defined peaks. **c** Associated periodic lattice distortions (PLDs) move tantalum nuclei (black spots) along the charge density gradient. Inset) Simulated diffraction shows sharp superlattice peaks decorating Bragg peaks. **d** Schematic representation of ordered IC-CDW in endotaxial polytype heterostructure. Mono- or few layers of endotaxially protected Oc-TaS$_2$ hosts 2D ordered IC-CDWs.

**e** Schematic representation of hexatic IC-CDW. The CDW phase is quasi-2D with non-trivial interlayer interactions, and hexatically disordered. **f** Charge density distribution is comparable to hexatically disordered crystal lattice. Inset) Structure factor reveals azimuthally diffused peaks—characteristics of hexatic phases. **g** Associated lattice distortion of IC-CDW with (inset) Fourier transform showing azimuthally blurred superlattice peaks while maintaining sharp Bragg peaks. **h** Schematic representation of hexatic IC-CDW in bulk 1T-TaS$_2$ where every layer hosts disordered IC-CDW.

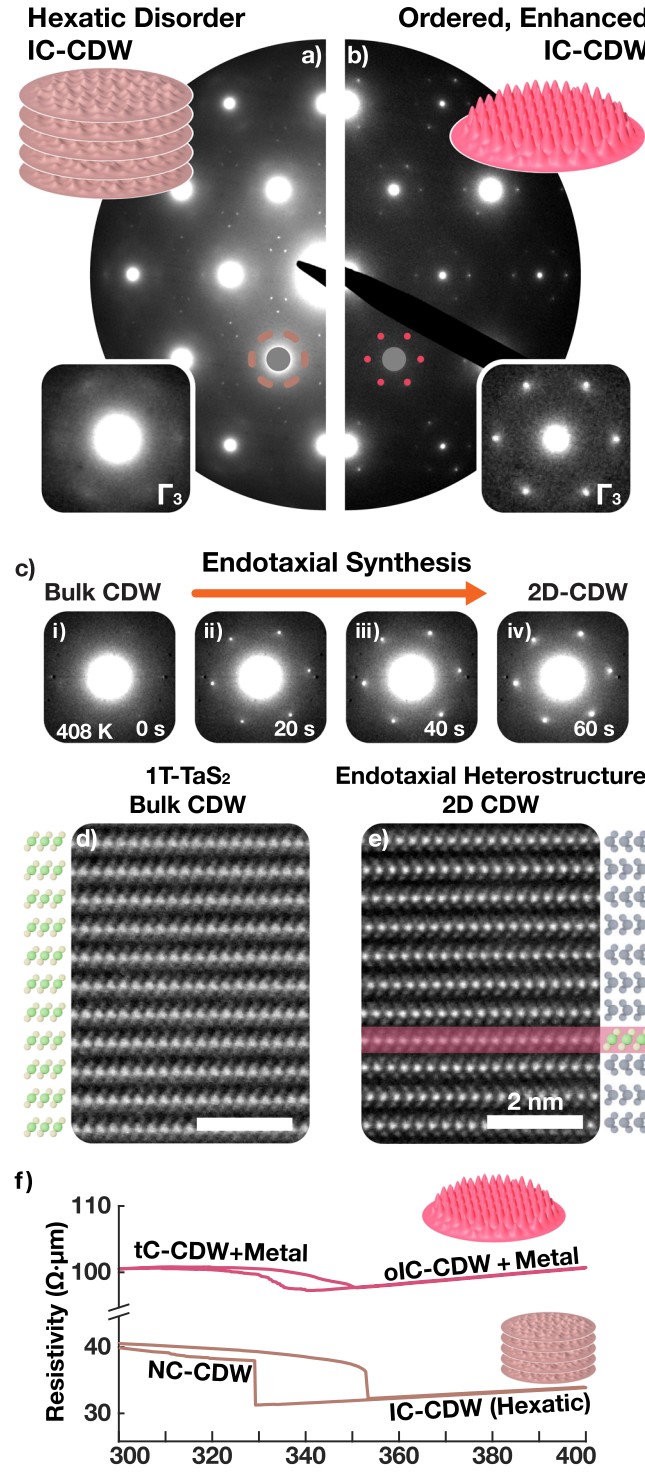

**Fig. 2 | Endotaxial polytype heterostructure of TaS₂. a** In bulk TaS₂, an IC-CDW phase emerges above 350 K, with azimuthally diffused superlattice peaks characteristic of hexatic disorder. **b** oIC-CDW in endotaxial polytype heterostructure has enhanced long-range order and amplitude. Superlattice peaks are well-defined, sharper and brighter. **c** Evolution of IC-CDW during the endotaxial synthesis. Atomic resolution cross-sectional HAADF-STEM of (**d**) bulk and (**e**) heat-treated TaSₓSe₂₋ₓ confirms polytypic transformation. After treatment, Pr layers encapsulate monolayers of Oc layers. Scale bar is 2 nm. A selenium-doped sample was imaged to enhance chalcogen visibility. **f** Resistivity vs. temperature measurement of bulk (brown) and thermally-treated (red) TaS₂ shows an increase in resistivity. In pristine sample, IC-CDW gives way to nearly commensurate (NC-) CDW around 350 K. In polytype heterostructure, twinned commensurate (tC-) CDW emerges at a similar temperature range.

Sharpening is especially highlighted in averaged third-order Bragg peaks ($\Gamma_3$, See Supplementary Fig. 2). The measured superlattice peaks of oIC-CDW are sharper both in azimuthal (by ~60%) and radial (by ~50%) directions when compared to the IC-CDW. Notably, the superlattice peak widths of the oIC phase is comparable to the peak widths of the principal Bragg peaks. Therefore, the oIC is a spatially coherent electronic crystal.

The oIC-CDW, a 2D charge-ordered state, is enhanced by at least one-hundred fold over previously reported bulk IC-CDWs. SAED uses a transmitted beam that measures the embedded oIC-CDWs and the metallic matrix. Diffracted superlattice peaks in oIC-CDWs have an integrated intensity over ten times stronger despite that the number of charge-ordered TaS₂ layers has been reduced to less than 10% of the material. Thus, endotaxial engineering improves not only the long-range order but also the charge order amplitude of the IC-CDW.

## Endotaxial polytype heterostructure of TaS₂

The oIC-CDW phase reported herein is stabilized by synthesizing endotaxial polytype heterostructures of TaS₂, where oIC-CDWs reside in monolayers of octahedrally coordinated (Oc-) TaS₂ embedded within prismatic (Pr-) TaS₂ matrix and one-to-one atomic registry (Fig. 2e). oIC phase occurs in isolated monolayers of Oc-TaS₂, but multiple instances of monolayer Oc-TaS₂ may be embedded in Pr-TaS₂ matrix—especially in thicker specimens (See Supplementary Fig. 3). Endotaxial polytype heterostructures are synthesized by heating 1T-TaS₂ at ~720 K for 15–30 min in an inert environment. Notably, 1T-TaS₂ is metastable and goes through Oc-to-Pr endotaxial layer-by-layer polytype transformation upon heating (≥620 K, See Supplementary Video). As the layer-by-layer polytype transitions are uncorrelated, the final system is not well described by bulk polytype symmetry groups—including the periodic 4-layer unit cell called 4Hb. In-situ SAEDs (Fig. 2c i–iv) were acquired at 20 s intervals at 408 K through the high-temperature conversion process (723 K). Within the first 20 seconds formation of metallic layers isolate 2D Oc layers with oIC CDWs that contribute sharp superlattice peaks in the SAED. These snapshots reveal sharpening of superlattice peaks—a clear indicator of enhanced CDW order. Cooling the sample midst transition stops the conversion and an interleaved polytype heterostructure is synthesized—confirmed by cross-sectional ADF-STEM.

Figure 2d, e shows atomic resolution micrographs of bulk 1T endotaxially converted to a polytype heterostructure. The atomic resolution images demonstrate endotaxial monolayer encapsulation of Oc-TaS₂ (Fig. 2e, highlighted red) in Pr-layers. The Pr-TaS₂ (bulk: 2H, 3R) are metallic above ~100 K. Previous work showed these metallic layers decouple CDWs out-of-plane and raise the critical temperature for commensurate quantum states (i.e., C-CDW) from ~200 K to ~350 K[36].

Surprisingly, the endotaxial polytype heterostructure stabilizes long-range order in IC-CDWs at elevated (≥350 K) temperatures. The oIC-CDW phase has a correlation length comparable to the crystal lattice, quantified by comparing widths of both superlattice and Bragg peaks from in-situ selected area electron diffraction patterns (SA aperture: 850 nm diameter). The FWHM (full width at half maximum) of CDW peaks are 10% larger than the Bragg peaks (See Supplementary Fig. 4). This indicates the CDW is relatively ordered (i.e., spatially coherent) over the distances comparable to the parent atomic crystal (~10² nm).

This enhancement of long-range CDW order is accompanied by an increase of the resistivity of the TaS₂ (Fig. 2f). Figure 2f shows temperature vs. in-plane resistivity measurement of 1T (brown) and endotaxial (red) specimen. The resistivity of endotaxial TaS₂ is higher for IC-CDW phases (>358 K), despite having many metallic layers introduced to the system. We observe an increase in resistance anisotropy (out-of-plane vs. in-plane) when the TaS₂ is thermally treated into endotaxial heterostructure (Supplementary Fig. 5). In Fig. 2f, the

flake was electronically contacted from the bottom (Supplementary Fig. 6). In this geometry, a sufficiently strong insulating layer can reduce current in conducting layers above. This implies that oIC-CDWs have higher resistivity than bulk 1T-TaS$_2$.

## Hexatic melting of IC-CDW

Creating the oIC-CDW provides an ordered charge lattice that can be hexatically melted upon further heating. Hexatic melting is a uniquely 2D process wherein a crystal melts in two stages through the creation of dislocations and disclinations[35,37–40]. During this process, the reciprocal space structure continuously evolves. Initially, at lower temperatures (c.a. 350 K), the oIC phase is an ordered charge crystal with well-defined peaks in reciprocal space (Fig. 3c). As temperature rises, the CDW peaks continuously blur azimuthally as the density of dislocations and disclinations increases (Fig. 3d, e). Azimuthal blurring of the reciprocal lattice is characteristic of hexatic phases and reflects the loss of translational symmetry while maintaining some orientational order[34]. Eventually, at higher temperatures (c.a. 570 K), the hexatic crystal completely dissociates into an amorphous liquid state with a ring-like structure factor. Figure 3c–e are generated using a phenomenological Monte Carlo simulation wherein displacement of the CDW charge centers follow a temperature-dependent Maxwell-Boltzmann probability distribution (See Methods). Here, the incommensurate CDW hexatically melts while the underlying atomic lattice remains unchanged—in diffraction this corresponds to a blurring of CDW superlattice peaks and preservation of Bragg peaks.

During the hexatic melting of oIC-CDWs, superlattice peaks increasingly blur as temperature is raised—clearly visible in in-situ SAED at Fig. 3a-i) 408 K, Fig. 3a-ii) 523 K, and Fig. 3a-iii) 573 K. The blurring is anisotropic and more prominent along azimuthal directions as expected for hexatic phases (See Supplementary Fig. 7). The CDW peaks are quantified throughout the melting process in Fig. 3b. Azimuthal peak width (Fig. 3b, blue-triangles) increases continuously with temperature; roughly doubling when raised from 410 K to 570 K. Around 520 K the oIC has melted into a state that resembles the well-known IC-CDW for bulk TaS$_2$. This CDW melting process is reversible and peaks sharpen when temperature is decreased. Notably, Bragg peaks do not show appreciable changes indicating only the electronic crystal is melting, not the TaS$_2$ atomic crystal.

Although the CDW melting process appears hexatic, it is distinct from familiar liquid crystals, silica spheres, or atomic crystals, wherein the amplitude of the order parameter does not change. Here, quantitative analysis of the superlattice peak intensities (Fig. 3a-red) reveals the charge density wave amplitude decreases with temperature. This is expected as topological defects in CDWs (dislocations and disclinations) have locally divergent strain with elastic energy cost that forces a local amplitude collapse. These local CDW amplitude collapses have been observed at the center of topological defects in the 3D charge ordering of manganites[1].

## The CDW phase diagram for octahedral TaS$_2$

Endotaxial synthesis of octahedrally coordinated TaS$_2$ allows access to new phases of matter and construction of a phase diagram for CDWs using temperature (T) and disorder ($\sigma$). The CDW phase diagram for 1T-TaS$_2$ is shown in Fig. 4. 1T-TaS$_2$ exists with native disorder and the ordered, commensurate phase (C-CDW, Fig. 4g) is only observed at low temperatures. At room temperature, the CDW is a partially-ordered NC phase (Fig. 4f) that enters the hexatic IC phase upon heating (Fig. 4e). At high temperatures or high disorder, CDWs degrade or vanish. Unlike lower temperature (<350 K) CDW phases in TaS$_2$ with chirality (broken mirror symmetry)[36,41,42], in both hexatic and ordered IC-CDW phases, this symmetry is not broken. The high disorder regime was historically achieved by substituting tantalum ions with other metal species (e.g., Ti, Nb) or by forcing intercalates within the van der Waals gap[24]. At room temperature, mild substitution of

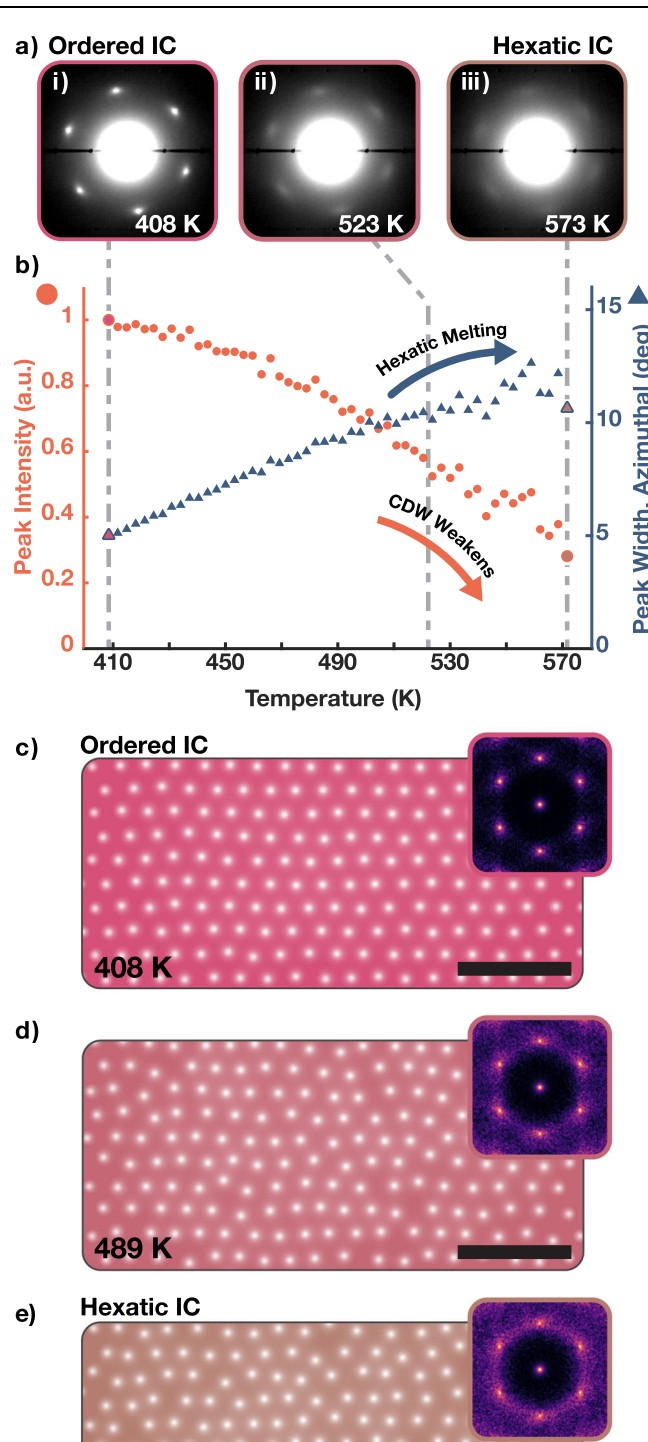

**Fig. 3 | Hexatic Melting of IC-CDWs. a** Averaged in-situ SAED patterns showing oIC-CDW superlattice peaks in endotaxial heterostructure. i–iii) As temperature increases (408 K, 523 K, 573 K), superlattice peaks continuously blur along azimuthal direction. **b** Quantification of superlattice peak profile. **b**-blue) Azimuthal width of the peak continuously increases with temperature—a key feature in hexatic melting process. **b**-red) Integrated superlattice peak intensity of oIC phase monotonically decays as temperature increases despite the increase in peak width; CDW is weakening. **c**–**e** Monte Carlo simulation of 2D Lennard-Jones crystal with increasing temperatures. This represents the charge density distribution. As temperature increases, the crystal progressively disorders with increasing numbers of disclinations and dislocations. Insets) Structure factor of the simulated crystals. Sixfold symmetry is apparent. As temperature increases, peaks diffuse prominently along azimuthal direction.

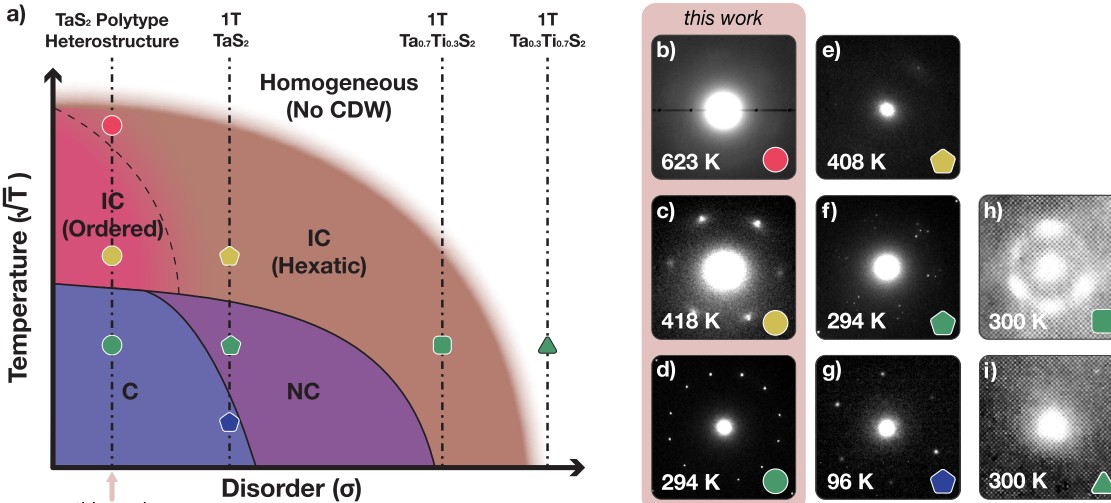

**Fig. 4 | Phase diagram of octahedrally coordinated TaS₂. a** Schematic temperature vs. disorder phase diagram of octahedrally coordinated $TaS_2$. As extrinsic disorder ($\sigma$) decreases, more ordered CDW phases are stabilized. At room temperature, polytype heterostructures with low disorder stabilize C-CDW (**d**) instead of NC-CDW (**f**), and long-range ordered IC-CDW (**c**) phase instead of hexatically disordered IC-CDW (**e**). Furthermore, it stabilizes CDWs (**b**) at higher temperatures than bulk $1T\text{-}TaS_2$ can ($T_{CDW} \cong 540\,K$[52]). Substitutional disorder, on the other hand, destroys long-range order and hexatic IC-CDW is stable at room temperature (**h**) and leads to complete destruction of CDW eventually (**i**). **b–i** Electron diffraction patterns showing superlattice peaks around a single Bragg peak reveal the charge ordering states. **h**, **i** are adapted from Wilson et al.[24].

titanium ($1T\text{-}Ta_{0.7}Ti_{0.3}S_2$) drives the system into hexatic-IC CDW states (Fig. 4h), and as more titanium is substituted ($1T\text{-}Ta_{0.3}Ti_{0.7}S_2$) CDW vanishes completely (Fig. 4i).

The low disorder regime, now accessible by endotaxial engineering, provides room temperature ordered C-CDWs[36] and a novel ordered IC-CDW at higher temperatures. Notably with low-disorder, the C to IC transition is direct and the NC phase does not appear. The IC phase is ordered, but the CDW can be continuously melted into a disordered hexatic-IC phase (as described in Fig. 3). The structure and boundaries of the CDW diagram are drawn with consistency with theoretical considerations for 2D symmetry breakings[18]. The curvature of the phase diagram boundaries is qualitative but has a physical functional form: energy is proportional to temperature and disorder squared ($E \propto T, \sigma^2$). Therefore, with the choice of ($\sqrt{T}, \sigma$) axis, the equi-energy surface is elliptical. Notably, this phase diagram shares similar features with nematic-CDWS[19,43] and 2D colloidal particles[44].

Notably, CDWs in endotaxial $TaS_2$ are two dimensional and the oIC phase has enhanced order despite the 3D to 2D dimensionality reduction. In bulk $1T\text{-}TaS_2$ CDWs are quasi-2D with non-negligible out-of-plane interaction (Fig. 1h)[45–48]. Formation of endotaxial polytype heterostructures disrupts the out-of-plane interactions and CDWs reside in a protected 2D environment[36]. Stabilization of an ordered IC-CDW in 2D seemingly contradicts with Hohenberg-Mermin-Wagner theorem[14,15] and Imry-Ma argument[17] which state spontaneous symmetry breaking of continuous symmetry (e.g., IC-CDWs) is unstable at non-zero temperatures in 2D. While both principles do not prevent intermediate phases with short-range order, the 2D CDWs should be nonetheless more fragile to disorder[18]. An ordered IC phase can only emerge in ultra-clean environments. Here endotaxial synthesis protects CDW states by strain-free encapsulation in a chemically identical environment of metallic layers that shield disorder. The reduction of environmental disorder provided by endotaxy may prove essential to accessing ordered CDW states in low dimensions.

## Discussion

In summary, we demonstrate that endotaxial synthesis of clean interleaved polytypic heterostructures can stabilize fragile quantum phases such as ordered CDWs even at high temperatures. Here, we stabilize and enhance 2D charge density waves (both long-range order and amplitude) in an endotaxially confined monolayer of $1T\text{-}TaS_2$. Surprisingly, the low-dimensional symmetry breaking of an ordered incommensurate CDW (oIC-CDW) appears, suggesting the quantum states reside within minimal extrinsic disorder. By enhancing CDW order the hexatic nature of IC-CDWs is revealed. Experimental observation matches advanced simulation of electron diffraction of charge lattices to provide the real-space evolution of 2D CDW melting. Heating the oIC-CDW in-situ TEM above 400 K we see a reversible hexatic melting process in which disclinations and dislocations destroy long-range translational symmetry of the CDW while maintaining its orientational order. The CDW melts well before the underlying atomic crystal changes. In 2D, CDWs are expected to manifest through vestigial electronic hexaticity–a weak CDW with substantial defects and short-range order. The nature of vestigial phases in CDWs remains poorly understood with little direct evidence. From these results, a CDW phase diagram for $1T\text{-}TaS_2$ is created and consistent with the predicted emergence of vestigial quantum order.

## Methods

### Simulated diffraction of charge lattices with heating

Charge density waves are electronic modulations describable in reciprocal space by three wave vectors (so-called, triple q) or in real-space as local charges arranged into a hexagonal lattice. For a fully ordered system, the charge lattice is a perfect lattice (Fig. 1b left), and the structure factor (Fig. 1b left inset) is also a perfect lattice. Here, the periodicity is equal to the incommensurate CDW wave vector $\mathbf{q}_{IC}$ (or $\mathbf{a}_{IC}$ in real-space). Traditional CDW theory elegantly describes ordered (or slightly disordered) systems using sparse representation in reciprocal space for ordered systems. However, a real-space basis readily describes topological disorder (dislocations and disclinations) in a charge density wave. This becomes particularly critical for IC phase (>350 K) of $1T\text{-}TaS_2$, where diffraction studies reveal azimuthally diffused superlattice peaks[24] that we show to be consistent with topological disorder in CDWs. Describing disorder of CDW plays a critical role in simulating experimentally consistent diffraction patterns at high temperatures.

The hexatic melting of a real-space charge lattice is illustrated with phenomenological Monte Carlo simulations of the NPT ensemble

(constant particle count, temperature, and pressure). The displacement of charge centers in a CDW follows a Maxwell-Boltzmann probability distribution at different temperatures. The interaction energy between charge centers is calculated using a shifted Lennard Jones potential truncated at 18.7 Å. From these first principles, the likelihood of forming dislocations and disclinations in a CDW lattice increases with temperature.

Diffraction of the simulated CDWs is calculated from the corresponding PLD of a 1T-TaS$_2$ crystal. The displacements are small ($\lesssim$10 pm), but clearly manifest as superlattice peaks with distinctive intensity in SAED. Notably, the superlattice peak intensities become stronger at higher $|\mathbf{k}|$; this is distinguishable from chemically ordered superlattice peaks that decay as $|\mathbf{k}|$ increases[23]. In TaS$_2$, atoms displace toward the charge centers which is equivalent to a longitudinal displacement wave. Here, the displacement amplitude is proportional to the charge density gradient with a max displacement set at 7 pm. Electron diffraction is kinematically simulated under flat Ewald Sphere approximations using the Fourier transform of the displaced atomic lattice.

### Electron microscopy

In-situ SAED was performed on Thermofisher Scientific (TFS) Talos (operated at 200 keV, SA aperture 850 nm) with Protochips Fusion Select holder and Gatan OneView Camera. Cross-sectional HAADF-STEM images were taken on JEOL 3100R05 (300 keV, 22 mrad) from ~150 nm thick TaS$_2$ specimen flakes prepared on TFS Nova Nanolab DualBeam FIB/SEM.

TEM specimens were prepared by exfoliating bulk 1T-TaS$_2$ and 1T-TaS$_x$Se$_{2-x}$ crystals onto polydimethylsiloxane (PDMS) gel stamp. The sample was then transferred to TEM grids using home-built transfer stage. Silicon nitride membrane window TEM grid with 2 μm holes from Norcada and Porotochips Fusion Thermal E-chips. From optical contrast and CBED patterns, the samples (Fig. 1, 2) were estimated to be 20–50 nm thick[49,50].

### Synthesis and acquisition of bulk crystals

1T-TaS$_2$ for in-situ SAED measurements and electronic measurements was acquired from HQ Graphene. 1T-TaS$_x$Se$_{2-x}$ ($x \approx 1$) for cross-sectional HAADF-STEM measurements was grown by the chemical vapor transport method with iodine as a transport agent. Stoichiometric amounts of the raw materials, high-purity elements Ta, S, and Se, were mixed and heated at 1170 K for 4 days in an evacuated quartz tube. Then the obtained TaS$_x$Se$_{2-x}$ powders and iodine (density: 5 mg/cm$^3$) were sealed in another longer quartz tube, and heated for 10 days in a two-zone furnace, where the temperature of source zone and growth zone was fixed at 1220 K and 1120 K, respectively. A shiny mirror-like sample surface was obtained, confirming their high quality. All CDW characterization was done on 1T-TaS$_2$; Se-doped sample was used only for polytype characterization in cross-sectional HAADF-STEM (Fig. 2d, e).

### Endotaxial synthesis of oIC-CDW in TaS$_2$

Interleaved 2D TaS$_2$ polytypes were synthesized by heating 1T-TaS$_2$ to 720 K in high vacuum (<10$^{-7}$ Torr) or in an argon-purged glovebox[36]. 1T-TaS$_2$ was held at 720 K for ~10 min, then brought down to room temperature. Once the interleaved polytype is fully established, the oIC-CDW becomes stable electronic state above 350 K[51].

### Device fabrication and electronic measurement

For resistivity measurements, TaS$_2$ flakes were transferred using PDMS gel stamp method to pre-fabricated bottom contacts. The fabrication of bottom contacts is detailed in ref. 36. The flake was sculpted into rectangular bar (~11 μm × 15 μm) using TFS Nova Nanolab DualBeam FIB/SEM (See Supplementary Fig. 6). The thickness of the flake was determined by AFM. For anisotropy measurements, SiN$_x$ membrane with pre-patterned gold electrodes was transferred over the flake (See Supplementary Fig. 5).

Resistivity vs. temperature measurements were performed in a Quantum Design Dynacool PPMS using a standard sample puck and an external Keithley 2400 series source meter. The sample was adhered to the puck backplane with silver paint, and contacts were wire bonded to the puck channel pads using 50 μm Au wire. To ensure sample thermalization, a baffle rod with an Au-coated sealing disk hovering < 1 cm above the sample was inserted into the PPMS bore, and the heating and cooling rate was restricted to <2 K/min. 10 μA current was sourced for four wire measurements. The current/voltage limits were chosen to keep electric fields below 10 kV/cm to avoid sample breakdown, as well as to keep current densities below 10$^5$ A/cm$^2$ and prevent localized heating at low temperatures.

## Data availability

In-situ SAED and ADF-STEM data are available at https://doi.org/10.5281/zenodo.10476898.

## Code availability

Matlab script for simulating the charge lattice model and associated lattice distortions is available at https://doi.org/10.5281/zenodo.10472701.

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

## Acknowledgements

This work acknowledges the scientific mentorship and insight of Lena F. Kourkoutis—an admired leader in electron microscopy who passed during the preparation of this manuscript.

S.H.S. and R.H. acknowledge support from the U.S. Department of Energy, Basic Energy Sciences, under award DE-SC0024147. Experiments were conducted using the Michigan Center for Materials Characterization (MC2) with assistance from Tao Ma and Bobby Kerns. This work made use of the facilities supported by the National Science Foundation through the Materials Research Science and Engineering Center at the University of Michigan, Award No. DMR-2309029 and the Platform for the Accelerated Realization, Analysis, and Discovery of Interface Materials (PARADIM) at Cornell under Cooperative Agreement No. DMR-2039380. N.S. acknowledges additional support from the NSF GRFP under award number DGE-2139899. P.K. and J.H. gratefully acknowledge support from NSF MRSEC DMR-2011839. Y.L., W.J.L., and Y.P.S., thank the support from the National Key R&D Program (Grant Nos. 2022YFA1403203 and 2021YFA1600201), the National Natural Science Foundation of China (Grant No. U2032215, No. U1932217 and No. 12274412).

Figure 4 h, i were adapted from Wilson et al.[24] with permission from Taylor & Francis Ltd.

## Author contributions

S.H.S and R.H. conceived the charge lattice model and associated lattice distortions and linked them to diffraction of TaS$_2$. S.H.S., Y.M.G., N.S., L.F.K., and R.H. performed HAADF-STEM and in-situ TEM and interpreted electron microscopy data. S.H.S., J.M.S., and N.A. fabricated samples for electronic measurements. P.K., J.M.S., N.A., T.C., and J.T.H. performed and analyzed electronic measurements. S.H.S., I.E.B., R.H., and K.S. provided theoretical interpretation. S.H.S. and N.A. performed Monte-Carlo simulations. S.H.S., K.S., and R.H. created the phase diagram of octahedrally coordinated TaS$_2$. Y.L., W.L., and Y.P.S. synthesized 1T-TaS$_x$Se$_{2-x}$ crystal. S.H.S. and R.H. prepared the manuscript. All authors reviewed and edited the manuscript.

## Competing interests

The authors declare no competing interests.

## Additional information

**Supplementary information** The online version contains
supplementary material available at

Robert Hovden.

**Peer review information** *Nature Communications* thanks Zhihai Cheng,
and the other, anonymous, reviewers for their contribution to the peer
review of this work. A peer review file is available.

