## [Peer Review File · Nature Communications]

REVIEWER COMMENTS

Reviewer #1 (Remarks to the Author):

The manuscript “Endotaxial Stabilization of 2D Charge Density Waves with Long-range Order” by Sung et al. reports a stabilize ordered incommensurate charge density waves (oIC-CDW) in TaS₂ using in-situ SAED. The new oIC-CDW phase is realized in the confined monolayer of 1T-by synthesizing endotaxial polytype heterostructures of TaS₂. They clearly give a phase diagram for CDWs using temperature (T) and disorder (σ) upon heating via in-situ TEM, and phenomenally discussed based on the Monte Carlo simulations.

The findings are relatively interesting and novel in the polytype heterostructures. Therefore, in my opinion, the manuscript deserves to be considered to be published in Nature Communications, provided that the following comments are addressed and taken into account in the following version of the draft.

1. The white dot in Figure 1b and black dot in Figure 1c is not defined. Does the black dot represent a Ta atom? The associated periodic lattice distortion (PLD) moving the tantalum nucleus along the charge density gradient cannot be clearly seen in the atomic lattice.
2. The authors use in-situ selected area electron diffraction (SAED) to investigate CDW states and the Fourier transform to determine whether the ICCDW is ordered-IC or hexatic-IC. Are there other experimental methods that can be used to verify the existence and nature of ordered-IC phase? The authors could provide more explanation and discussion to clarify the soundness of their choice of experimental method.
3. Although the authors mention that the oIC-CDW phase can be stabilized by synthesizing endotaxial polytype heterostructures of TaS₂, they do not provide an adequate explanation of why endotaxial polytype heterostructures can achieve an ordered IC-CDW phase?
4. The authors state that 1T-TaS₂ goes through Oc-to-Pr endotaxial layer-by-layer polytype transformation upon heating. Can they explain why this transition is layer-by-layer and what experimental evidence can be used to demonstrate this? How can it be confirmed that only a single layer of Oc-TaS₂ remains after the transformation? Any XRD measurements have been performed to confirm this?
5. In Figure 2c, the authors show the in-situ SAED images 20 seconds intervals through the high temperature conversion process. However, sharp superlattice peaks can already be seen in Fig. 2c ii, is the CDW already partially ordered in this state? What is its corresponding atomic resolution image like? Are there several layers of untransformed Oc-TaS₂ present?
7. It is somewhat surprising that the resistivity of oIC-CDWs is higher than hexatic-IC in 1T-TaS₂. Can the authors rationalize this?

8. It is not clear why the C to IC transition is direct and the NC phase does not appear. Could the authors elaborate more on this?
9. The authors give the phase diagrams for temperature (T) and disorder (σ) in Fig. 4. How did the authors determine the boundaries of each phase in the phase diagrams? And the authors mention the consistency of the phase diagram boundaries with consistency to hexatic melting of 2D colloidal particles under temperature and disorder as well as nematic CDWs. However, the authors do not provide sufficient experimental evidence or detailed explanations to support the plausibility of this description?
10. Could the authors briefly discuss the difference between the endotaxial polytype heterostructure and 4Hb-TaS₂. According to the crystal structure of 4Hb-TaS₂, there is also the ordered confined monolayer of 1T-TaS₂ between the 1H-TaS₂.
11. The chirality is also an important feature in the CDW phase of 1T-TaS₂. The authors also discussed the possible coexistent chiral-domains in the endotaxial polytype heterostructures of TaS₂.

Reviewer #2 (Remarks to the Author):

In this work, Song and coauthors demonstrate that heat treating TaS₂ produces a phase change such that a CDW is encapsulated inside metallic layers (hence “endo” in the title). That process seems to stabilize the CDW phase in the layer to unexpectedly high temperatures. It is a clever idea and demonstrates careful work.

I like the figures. I like the presentation and discussion. I felt I learned something about this material.

Overall the work by Song et al is very well written and extremely well presented. After considering the manuscript I don't have any notable criticism. In my opinion it can be accepted as is.

Reviewer #3 (Remarks to the Author):

The manuscript "Endotaxial Stabilization of 2D Charge Density Waves with Long-range Order" by Sung et al. describes TEM measurements of CDW in 1T-TaS₂ and endotaxial polytype of TaS₂. The main conclusion of the article is that in endotaxial TaS₂ the incommensurate CDW has long range

order that is much higher than 1T-TaS₂. The reason for this increase in ordering is that the CDW layers are decoupled with metallic layers in endotaxial case.

The subject is timely and interesting. The data is presented in accessible form. The conclusions that are provided are not very well supported by the data and that is why I would like to ask the Authors to revise the manuscript taking into account the comments below.

1. The authors claim that the enhanced long-range ordering of the IC CDW in endotaxial polytype of TaS₂ results in a higher in-plane resistivity as larger electron density is localized on the lattice ("This enhancement of long-range CDW order is accompanied by a marked increase of the in-plane resistivity of the IC phase (Fig. 2f).") On the other hand, there are many metallic" layers between the two nearest oIC- CDW layers.

This claim is unsubstantiated and contrary to what one would expect if one compares a system with many semiconducting layers to the system with many metallic layers and few insulating layers in-between. One would naturally expect that in-plane resistivity will be shunted by the metallic layers and one should get lower resistance in the case of the endotaxial TaS₂. This is not what experiment shows.

In order for this claim to be substantiated with data one would need to show the anisotropy in conductivity. Since the oIC-CDW layers are essentially insulating the anisotropy should be much higher in endotaxial case with oIC-CDW than in 1T-TaS₂ IC-CDW case.

Otherwise, it is not possible to reconcile the argument that oIC CDW is localized in few layers sandwiched between metallic planes as described in the text. All the claims about anisotropy in conductivity in the text are unsubstantiated by any results since there is no data on this.

2. The authors do not calculate the CDW correlation length in the plane and along c-axis. It would be good to give real numbers as they can be extracted from the existing data.

3. What are thicknesses of the TEM-imaged samples both in case of e-beam along the c-axis (SAED) and ab-plane (ADF-STEM fig.2(d,e))? It would be important to provide more details about the experimental parameters. In addition, it is not clear how many CDW carrying layers are in the case of endotaxial TaS₂? In the text seems that there is only one CDW carrying layer when the sample is imaged by SAED?

Clearly in Fig.2e there is only one layer, but it is not clear what is the periodicity of this layer along c-axis (how many metallic layers are in-between two CDW layers and how ordered is the period (does the number of metallic layers vary))?

4. It would be necessary to show the CDW peak profiles (azimuthal and radial) and how they change with temperature as the hexatic melting takes place. This can be done in supplementary info. The reader would always be curious to compare the peak shape and peak area as the melting proceeds.

5. It would be desirable (but not essential) to substantiate the claims about charge transfer between planes when transitioning from 1T to endotaxial polytype by some DFT calculations. One might learn from that the level of charge localization and compare it with the data. PDOS would also be helpful.

Overall, this is an interesting manuscript that needs to be revised in order to be considered for publication.

Robert Hovden, PhD
Associate Professor
Materials Science & Eng.
University of Michigan
hovden@umich.edu
hovdenlab.com
c. 770-265-4042

Response to Reviewers

Reviewer #1:

The manuscript “Endotaxial Stabilization of 2D Charge Density Waves with Long-range Order” by Sung et al. reports a stabilize ordered incommensurate charge density waves (oIC-CDW) in TaS₂ using in-situ SAED. The new oIC-CDW phase is realized in the confined monolayer of 1T-by synthesizing endotaxial polytype heterostructures of TaS₂. They clearly give a phase diagram for CDWs using temperature (T) and disorder (σ) upon heating via in-situ TEM, and phenomenally discussed based on the Monte Carlo simulations.

The findings are relatively interesting and novel in the polytype heterostructures. Therefore, in my opinion, the manuscript deserves to be considered to be published in Nature Communications, provided that the following comments are addressed and taken into account in the following version of the draft.

We thank the reviewer for positive remarks.

1. The white dot in Figure 1b and black dot in Figure 1c is not defined. Does the black dot represent a Ta atom? The associated periodic lattice distortion (PLD) moving the tantalum nucleus along the charge density gradient cannot be clearly seen in the atomic lattice.

We have now defined the black and white dots in Figure 1b & c. The caption for Figure 1 now reads “b) Ordered IC-CDW illustrated as a crystalline charge-density lattice. Here white spots represent charge centers. c) Associated periodic lattice distortions (PLDs) move tantalum nuclei (black spots) along the charge density gradient.”

Supplementary Figure S1 has been added to show detail in the atomic lattice and the direction of motion along the charge gradient. On page 2 in the text body we now state. “charge maximum to form periodic clusters of atoms (Fig. 1c, see also Supplementary Figure S1).”

2. The authors use in-situ selected area electron diffraction (SAED) to investigate CDW states and the Fourier transform to determine whether the ICCDW is ordered-IC or hexatic-IC. Are there other experimental methods that can be used to verify the existence and nature of ordered-IC phase? The authors could provide more explanation and discussion to clarify the soundness of their choice of experimental method.

Robert Hovden, PhD
Associate Professor
Materials Science & Eng.
University of Michigan
hovden@umich.edu
hovdenlab.com
c. 770-265-4042

These ordered endotaxial states are embedded within a bulk matrix and require measurements that transmit through all layers. On page 3, para 1 we now state “SAED uses a transmitted beam that measures the embedded oIC-CDWs and the metallic matrix.” Although STM is capable of measuring CDWs, it is surface sensitive and would miss embedded layers. High energy electron diffraction is a transmission technique with a large (flat) Ewald sphere that measures out to several Bragg peaks and “is sensitive to subtle picoscale distortions and making it a popular choice for characterization of CDW/PLDs (page 2 para 2)”.

3. Although the authors mention that the oIC-CDW phase can be stabilized by synthesizing endotaxial polytype heterostructures of TaS₂, they do not provide an adequate explanation of why endotaxial polytype heterostructures can achieve an ordered IC-CDW phase?

This reproducible experimental work demonstrates that endotaxial synthesis of clean interleaved polytypic heterostructures stabilizes ordered CDW phases such as ordered incommensurate (oIC) CDWs even at high temperatures. By restoring in-plane order of the incommensurate phase we are able to subsequently melt the oIC phase into a level of disorder that is consistent with the well-known bulk IC-CDW. Nie, Tarjus, and Kivelson provide a theory for the stability of CDW phases in the presence of extrinsic disorder—showing that low disorder regimes are essential to stabilizing ordered incommensurate states.

On page 5 para 5, we have expanded our hypothesis as to why endotaxial polytype heterostructures can achieve ordered CDWs. “... the 2D CDWs should be none-the-less more fragile to disorder (Nie et al). An ordered IC phase can only emerge in ultra-clean environments. Here endotaxial synthesis protects CDW states by strain-free encapsulation in a chemically identical environment of metallic layers that shield disorder. The reduction of environmental disorder provided by endotaxy may prove essential to accessing ordered CDW states in low dimensions.”

4. The authors state that 1T-TaS₂ goes through Oc-to-Pr endotaxial layer-by-layer polytype transformation upon heating. Can they explain why this transition is layer-by-layer and what experimental evidence can be used to demonstrate this?

We now include an additional experimental video (SI Video) that shows the layer by layer transformation using an in situ heating of the specimen within the STEM. This experiment replicates the observation of endotaxy polytypism [R1]. Page 3 para 3 now reads “...endotaxial layer-by-layer polytype transformation upon heating (≥ 620 K, See Supplementary Video).”

Robert Hovden, PhD
Associate Professor
Materials Science & Eng.
University of Michigan
hovden@umich.edu
hovdenlab.com
c. 770-265-4042

[R1] Sung et al., Nat. Commun. 13, 413 (2022)

How can it be confirmed that only a single layer of Oc-TaS₂ remains after the transformation?

On page 3 para 3 we now clarify “oIC phase occurs in isolated monolayers of Oc-TaS₂, but multiple instances of monolayer Oc-TaS₂ may be embedded in Pr-TaS₂ matrix—especially in thicker specimens (See Supplementary Figure S3)”. We’ve added a larger field of view cross-sectional HAADF-STEM of endotaxial TaS₂ as Supplementary Figure S3. We appreciate the reviewer for this critical comment.

Any XRD measurements have been performed to confirm this?

Structural X-ray diffraction analysis will provide similar information compared to SAED. We have not done any XRD measurements.

5. In Figure 2c, the authors show the in-situ SAED images 20 seconds intervals through the high temperature conversion process. However, sharp superlattice peaks can already be seen in Fig. 2c ii, is the CDW already partially ordered in this state? What is its corresponding atomic resolution image like? Are there several layers of untransformed Oc-TaS₂ present?

We thank the reviewer for needed clarity here. On page 3, para 3 we now clarify what is happening in Fig. 2c, “Within the first 20 seconds formation of metallic layers isolate 2D Oc layers with oIC CDWs that contribute sharp superlattice peaks in the SAED.”

7. It is somewhat surprising that the resistivity of oIC-CDWs is higher than hexatic-IC in 1T-TaS₂. Can the authors rationalize this?

Here a mixture of CDW and metallic TaS₂ layers are measured in a 4-point bottom contact geometry. We agree that the introduction of metallic layers will shunt the in-plane resistivity. However, the total resistance can increase by introducing a strong insulator (oIC-CDW) within metallic layers nearer to the bottom contacts.

On page 4, para 2, we have added clarification “In this geometry, a sufficiently strong insulating layer can reduce current in conducting layers above. This implies that oIC-CDWs have a higher resistivity than bulk 1T-TaS₂”

Robert Hovden, PhD
Associate Professor
Materials Science & Eng.
University of Michigan
hovden@umich.edu
hovdenlab.com
c. 770-265-4042

We have conducted additional measurement of resistance anisotropy. On page 4, para 2 we state, “We observe an increase in resistance anisotropy (out-of-plane vs. in-plane) when TaS₂ is thermally treated into endotaxial heterostructure (Supplementary Figure S5).” Here, multiple top and bottom contacts of known geometry provide in-plane and out-of-plane resistivity. Beneath the top contacts, a SiN membrane was used to prevent sidewall contact.

8. It is not clear why the C to IC transition is direct and the NC phase does not appear. Could the authors elaborate more on this?

In this work, we focus on the emergence and melting of oIC and IC phases. Disappearance of NC-CDW phase was discussed in [R1]. In that work, it is hypothesized that “each 2D 1T-TaS₂ CDW is in its native chemical, endotaxial, and unstrained environment. Impurity potentials that pin CDWs and break spatial coherence are mitigated by adjacent metallic Pr-layers. For C-CDWs (in 2D and above) in the presence of sufficiently weak disorder, the charge order remains stable. Additionally, isolating monolayers of 1T-TaS₂ ensures an odd number of electrons per unit cell and elongates the Fermi surface out-of-plane—both expected to reduce the electronic energy.” Stabilizing the long-range ordered C phase quenches out the NC phase.

[R1] Sung et al., Nat. Commun. 13, 413 (2022)

9. The authors give the phase diagrams for temperature (T) and disorder (σ) in Fig. 4. How did the authors determine the boundaries of each phase in the phase diagrams? And the authors mention the consistency of the phase diagram boundaries with consistency to hexatic melting of 2D colloidal particles under temperature and disorder as well as nematic CDWs. However, the authors do not provide sufficient experimental evidence or detailed explanations to support the plausibility of this description?

We appreciate the reviewer for the needed discussion. On second paragraph of page 5, the last sentence now reads “The structure and boundaries of the CDW diagram are drawn with consistency with theoretical considerations for 2D symmetry breakings [18]. Notably, this phase diagram shares similar features with nematic-CDWs [19,41] and 2D colloidal particles [42]. The curvature of the phase boundaries is qualitative but has a physical functional form: energy is proportional to temperature and disorder squared ($E \propto T, \sigma^2$). Therefore, with the choice of \sqrt{T}, σ axis, the equi-energy surface is elliptical.”

Here we took inspiration from Kivelson et al. 's phase diagram and presented a schematic phase diagram based on Kivelson's vestigial order theory and experimental diffraction data. As Kivelson et

Robert Hovden, PhD
Associate Professor
Materials Science & Eng.
University of Michigan
hovden@umich.edu
hovdenlab.com
c. 770-265-4042

al. pointed out, the disorder in the low dimensional CDW system is extremely important but was often overlooked. In their manuscript, single Q (“nematic IC-CDW”) was studied but the argument is more or less same: long-range order in incommensurate CDW is impossible to achieve at finite temperature under any disorder, instead the vestigial order (i.e., nematic or hexatic) emerges.

10. Could the authors briefly discuss the difference between the endotaxial polytype heterostructure and 4Hb-TaS₂. According to the crystal structure of 4Hb-TaS₂, there is also the ordered confined monolayer of 1T-TaS₂ between the 1H-TaS₂.

The clearest explanation of our system is to describe it as an uncorrelated and sparse interleaving of octahedral layers within many prismatic layers. “As the layer-by-layer polytype transitions are uncorrelated, the final system is not well described by bulk polytype symmetry groups—including the periodic 4-layer unit cell called 4Hb.” (now added on page 3) The key takeaway is that the endotaxial sample is not a bulk 4Hb-TaS₂ sample and this controllable synthesis allows us to understand behavior in individual layers, polytype interfaces, and build devices around a tunable number of 2D CDW.

11. The chirality is also an important feature in the CDW phase of 1T-TaS₂. The authors also discussed the possible coexistent chiral-domains in the endotaxial polytype heterostructures of TaS₂.

As reviewer pointed out chirality can be a relevant CDW feature. We now clue the reader into considering the role of chirality in 1T-TaS₂. On page 5, para 2 we now state, “Unlike lower temperature (< 350 K) CDW phases in TaS₂ with chirality (here, broken mirror symmetry) [35 , 41, 42], in both hexatic and ordered IC-CDW phases this symmetry is not broken.”

[35] S. H. Sung et al., Nat. Commun. 13, 413 (2022)

[41] S. Husremović et al., Nat. Commun. 14, 6031 (2023)

[42] Y. Zhao et al., Nat. Commun. 14, 2223 (2023)

Robert Hovden, PhD
Associate Professor
Materials Science & Eng.
University of Michigan
hovden@umich.edu
hovdenlab.com
c. 770-265-4042

Reviewer #2:

In this work, Song and coauthors demonstrate that heat treating TaS₂ produces a phase change such that a CDW is encapsulated inside metallic layers (hence “endo” in the title). That process seems to stabilize the CDW phase in the layer to unexpectedly high temperatures. It is a clever idea and demonstrates careful work.

I like the figures. I like the presentation and discussion. I felt I learned something about this material.

Overall the work by Song et al is very well written and extremely well presented. After considering the manuscript I don't have any notable criticism. In my opinion it can be accepted as is.

We thank reviewer 2 for the positive remarks. Substantial effort was put into this work and its presentation.

Robert Hovden, PhD
Associate Professor
Materials Science & Eng.
University of Michigan
hovden@umich.edu
hovdenlab.com
c. 770-265-4042

Reviewer #3:

The manuscript "Endotaxial Stabilization of 2D Charge Density Waves with Long-range Order" by Sung et al. describes TEM measurements of CDW in 1T-TaS₂ and endotaxial polytype of TaS₂. The main conclusion of the article is the in endotaxial TaS₂ the incommensurate CDW has long range order that is much higher than 1T-TaS₂. The reason for this increase in ordering is that the CDW layers are decoupled with metallic layers in endotaxial case.

The subject is timely and interesting. The data is presented in accessible form. The conclusions that are provided are not very well supported by the data and that is why I would like to ask the Authors to revise the manuscript taking into account the comments below.

We appreciate the reviewer's overall positive remarks.

1. The authors claim that the enhanced long-range ordering of the IC CDW in endotaxial polytype of TaS₂ results in a higher in-plane resistivity as larger electron density is localized on the lattice ("This enhancement of long-range CDW order is accompanied by a marked increase of the in-plane resistivity of the IC phase (Fig. 2f).") On the other hand, there are many metallic" layers between the two nearest oIC- CDW layers.

This claim is unsubstantiated and contrary to what one would expect if one compares a system with many semiconducting layers to the system with many metallic layers and few insulating layers in-between. One would naturally expect that in-plane resistivity will be shunted by the metallic layers and one should get lower resistance in the case of the endotaxial TaS₂. This is not what experiment shows.

In order for this claim to be substantiated with data one would need to show the anisotropy in conductivity. Since the oIC-CDW layers are essentially insulating the anisotropy should be much higher in endotaxial case with oIC-CDW than in 1T-TaS₂ IC-CDW case.

Otherwise, it is not possible to reconcile the argument that oIC CDW is localized in few layers sandwiched between metallic planes as described in the text. All the claims about anisotropy in conductivity in the text are unsubstantiated by any results since there is no data on this.

We agree that the introduction of metallic layers will shunt the in-plane resistivity. However, the total resistance can increase by introducing a strong insulator (oIC-CDW) within metallic layers nearer to the bottom contacts. Here a mixture of CDW and metallic TaS₂ layers are measured in a 4 point bottom contact geometry.

Robert Hovden, PhD
Associate Professor
Materials Science & Eng.
University of Michigan
hovden@umich.edu
hovdenlab.com
c. 770-265-4042

As requested, we have now added data showing an increase in anisotropic transport. On page 4, para 2 we state, “We observe an increase in resistance anisotropy (out-of-plane vs. in-plane) when TaS₂ is thermally treated into endotaxial heterostructure (Supplementary Figure S5).” Here, multiple top and bottom contacts of known geometry provide in-plane and out-of-plane resistivity at room temperature. Beneath the top contacts, a SiN membrane was used to prevent sidewall contact. The anisotropy increased by approximately 5 times.

On page 4, para 2, we have removed the word “in-plane” because both in-plane and out-of-plane fields may contribute to the measured resistance of the flake. We have added clarification “In this geometry, a sufficiently strong insulating layer can reduce current in conducting layers above. This implies that oIC-CDWs have a higher resistivity than bulk 1T-TaS₂”

2. The authors do not calculate the CDW correlation length in the plane and along c-axis. It would be good to give real numbers as they can be extracted from the existing data.

We now provide real numbers extracted from the existing diffraction data.

On page 3, para 5 we now state “The FWHM (full width at half maximum) of CDW peaks are 10% larger than the Bragg peaks (See Supplementary Figure S4)”. We have also added the word “in-plane”. In the Supplementary Figure S4 we also state, “Here is best to compare the CDW peak width relative to the Bragg peaks under the same imaging conditions since diffraction peak widths are determined by the crystal / CDW order, size of the diffracting region, and the microscope optics.” Out of plane, the CDW layers are uncorrelated (Supplementary Figure S3).

3. What are thicknesses of the TEM-imaged samples both in case of e-beam along the c-axis (SAED) and ab-plane (ADF-STEM fig.2(d,e))? It would be important to provide more details about the experimental parameters. In addition, it is not clear how many CDW carrying layers are in the case of endotaxial TaS₂? In the text seems that there is only one CDW carrying layer when the sample is imaged by SAED?

We appreciate the need for clarification here. In order to make this point clear, on page 3 para 3 we now state “oIC phase occurs in isolated monolayers of Oc-TaS₂, but multiple instances of monolayer Oc-TaS₂ may be embedded in Pr-TaS₂ matrix—especially in thicker specimens”. We’ve added a larger field of view cross-sectional HAADF-STEM of endotaxial TaS₂ as Supplementary Figure S3

Robert Hovden, PhD
Associate Professor
Materials Science & Eng.
University of Michigan
hovden@umich.edu
hovdenlab.com
c. 770-265-4042

On page 8, para 5 we now specify, “Cross-sectional HAADF-STEM images were taken on JEOL 3100R05 (300 keV, 22 mrad) from ~150nm thick specimen TaS₂ flakes prepared on TFS Nova Nanolab DualBeam FIB/SEM.”

Clearly in Fig.2e there is only one layer, but it is not clear what is the periodicity of this layer along c-axis (how many metallic layers are in-between two CDW layers and how ordered is the period (does the number of metallic layers vary))?

We’ve added a larger field of view cross-sectional HAADF-STEM of endotaxial TaS₂ as Supplementary Figure S3. This data shows Oc-TaS₂ layers makeup roughly ~5% of all layers. The isolated monolayers are aperiodic. The final number of Oc-TaS₂ can be tuned in the synthesis process, although getting exactly one monolayer may be challenging.

4. It would be necessary to show the CDW peak profiles (azimuthal and radial) and how they change with temperature as the hexatic melting takes place. This can be done in supplementary info. The reader would always be curious to compare the peak shape and peak area as the melting proceeds.

We agree that providing the peak shape change as the melting proceeds is relevant and of interest to the reader. We have added Supplementary Figure S4 which provides analysis on the azimuthal and radial changes to CDW peak profiles. Page 4 para 4 now reads “...The blurring is anisotropic and more prominent along azimuthal directions as expected for hexatic phases (See Supplementary Figure S7)”

Robert Hovden, PhD
Associate Professor
Materials Science & Eng.
University of Michigan
hovden@umich.edu
hovdenlab.com
c. 770-265-4042

Fig. S7 | Thermal Evolution of oIC-CDW Superlattice Peaks. Intensity profile of superlattice peak in oIC-CDW phase along a) radial and b) azimuthal direction as temperature increases from 408.3 K (bottom, black) to 571.5 K (top, red). Continuous peak broadening and amplitude reduction is apparent.

5. It would be desirable (but not essential) to substantiate the claims about charge transfer between planes when transitioning from 1T to endotaxial polytype by some DFT calculations. One might learn from that the level of charge localization and compare it with the data. PDOS would also be helpful.

We agree that advanced calculations on charge transfer between planes and multi-body physics is worthwhile research. However, to do these calculations requires care and effort beyond the scope of this work. It may elucidate the underlying mechanisms of electronic screening or reveal additional CDW enhancements that occur at an endotaxial polytype interface.

Overall, this is an interesting manuscript that needs to be revised in order to be considered for publication.

REVIEWERS' COMMENTS

Reviewer #1 (Remarks to the Author):

The manuscript "Endotaxial Stabilization of 2D Charge Density Waves with Long-range Order" by Sung et al. stabilize ordered incommensurate charge density waves (oIC-CDW) in TaS₂. The oIC-CDW phase is realized by synthesizing endotaxial polytype heterostructures of TaS₂. The clearly give a phase diagram for CDWs using temperature (T) and disorder (σ) upon heating via in-situ TEM. The findings are interesting and novel, making it in principle appropriate as a letter.

Reviewer #3 (Remarks to the Author):

The authors all the issues. I have no further remarks and the paper can proceed to publication.